# Risk Factors of Urrets-Zavalia Syndrome after Penetrating Keratoplasty

**DOI:** 10.3390/jcm11051175

**Published:** 2022-02-22

**Authors:** Ping Wang, Qingqin Gao, Guanyu Su, Wei Wang, Lingjuan Xu, Guigang Li

**Affiliations:** 1Department of Ophthalmology, Eye Institute of China Three Gorges University, Affiliated Renhe Hospital of China Three Gorges University, Yichang 443000, China; pingwang1130@163.com (P.W.); gqq0226@163.com (Q.G.); 2Department of Ophthalmology, Tongji Hospital, Tongji Medical College, Huazhong University of Science & Technology, Wuhan 430030, China; sgy804408@163.com (G.S.); wwnissan@163.com (W.W.); xlj_doc@163.com (L.X.)

**Keywords:** Urrets-Zavalia syndrome, penetrating keratoplasty, keratoconus, ocular hypertension

## Abstract

Objective: The objective of this study was to analyze the clinical features and risk factors of Urrets-Zavalia syndrome (UZS) after penetrating keratoplasty (PKP). Methods: The medical records of 152 patients who underwent PKP at the Department of Ophthalmology, Tongji Hospital, between January 2014 and December 2016 were retrospectively reviewed. UZS was diagnosed based on pre- and post-operative pupillary findings. The relationships among the primary disease, postoperative intraocular pressure (IOP), and the incidence of UZS were statistically analyzed. The pupillary changes during the follow-up period were studied. Results: Among the 152 included patients, 23 were diagnosed with UZS, with an incidence of 15.13%. The primary diseases of the UZS patients were keratoconus (eight cases, 34.78%), viral keratitis (six cases, 26.08%), leukoma (four cases, 17.39%), fungal corneal ulcer (two cases, 8.70%), corneal endothelial decompensation (two cases, 8.70%), and corneal degeneration (one case, 4.35%). The incidence of UZS in keratoconus patients was higher than that in patients with fungal corneal ulcer (42.11% versus 6.25%, *p* = 0.003); In addition, the transient postoperative high IOP was not significantly related to the incidence of UZS in keratoconus patients in our study (*p* = 0.319). Twenty-one patients with UZS were followed up for >6 months, seven of whom (33.33%) recovered spontaneously (within the range of 48 days to 1.5 years). Conclusion: In our study, the incidence of UZS after PKP was 15.13%, and 33.33% of these patients recovered spontaneously. UZS may be more likely to occur in patients with keratoconus. Postoperative transient high IOP may increase the incidence of UZS after PKP for keratoconus.

## 1. Introduction

Since Urrets-Zavalia syndrome (UZS) was first reported in 1963 [1], it has been defined as a fixed wide pupil after intraocular surgeries that has not been affected by pupil-dilation medicine—including atropine and topicamide—or clear etiology, including preoperative uveitis, traumatic iris defects, preoperative pupillary abnormalities, gonioplasty, iridocorneal endothelial syndrome, excision of iris cysts etc. UZS has been widely observed in patients who undergo ophthalmological surgeries, such as penetrating keratoplasty (PKP) for keratoconus or corneal dystrophy [2]; cataract surgery [3]; intraocular lens implantation [4]; trabeculectomy [5]; deep lamellar keratoplasty [6,7]; Descemet-stripping endothelial keratoplasty (DSEK) [6]; Descemet membrane endothelial keratoplasty (DMEK) [8]; argon laser peripheral iridoplasty [9]; and goniotomy [10]. Due to the fixed and dilated pupils associated with UZS, in addition to discomfort under strong light, UZS affects patients’ best-corrected visual acuity and causes great inconvenience and pain. Wearing sunglasses or colored contact lenses are the only available treatments. Thus, preventing the occurrence of UZS and identifying and controlling the risk factors of UZS are more effective methods for improving patients’ quality of life.

Although the etiology of UZS has not yet been determined, iris ischemia is currently considered to be the main cause [1,11,12,13,14,15]. However, other factors affecting the incidence of UZS have also been identified, such as keratoconus [16,17]. Because the pathogenesis of UZS is unknown, the assessment and control of these related risk factors is hard to carry out.

Due to the large differences in economic development, sanitary conditions, and quality of the health of patients, the risk factors identified for UZS have varied dramatically in different studies, with the incidence ranging from 2.2% to 17.7% [12]. The aim of this study was to analyze the incidence rate of UZS with different primary diagnoses for PKP and other risk factors, such as high intraocular pressure during the perioperative period, which may be useful for the prevention and treatment of UZS.

## 2. Materials and Methods

A total of 244 patients who underwent PKP at the Department of Ophthalmology, Tongji Hospital, between January 2014 and December 2016 were retrospectively reviewed. Patients who had incomplete medical records, blurred anterior-segment photographs, had undergone canthorrhaphy, or were followed up for less than 3 months were excluded. This study followed the Declaration of Helsinki and was approved by the ethics committee of Tongji Hospital. The need to obtain informed consent was waived because of the retrospective nature of the study, and because the data were analyzed anonymously.

The data collected included age, sex, primary eye disease, anterior segment photography, postoperative intraocular pressure (IOP), and length of follow-up. UZS was diagnosed based on the changes in pupil diameter and light reflexes before and after surgery. The pupil was normal before surgery and enlarged after surgery. If a wide, fixed pupil was recorded 3 weeks after the final use of atropine or tropicamide—excluding clear etiology for pupil dilation, such as preoperative uveitis, traumatic iris defects, preoperative pupillary abnormalities, gonioplasty, iridocorneal endothelial syndrome, excision of iris cysts etc.—the diagnosis of postoperative UZS was made in this study. Depending on the size of the lesion, the donor’s corneal diameter ranged from 7.25 mm to 8.0 mm, while the recipient’s diameter was 0.25 mm smaller than the donor’s. The corneas were fixed with 16 stitches in a discontinuous suture or continuous suture. The operation time was around 40 to 50 min. In patients with keratoconus, bullous keratitis, and other diagnoses wherein the cornea is not so cloudy, pupil diameter could be measured before surgery; some other pupils could not be measured before surgery, namely those patients with a totally opaque cornea, and were instead measured during or after the cornea transplantation. The incidence of UZS in patients with different primary diseases was calculated. Patients with keratoconus were divided into two groups based on the presence or absence of transient high IOP after surgery, and the incidence of UZS in these two groups was compared. Spontaneous recovery of pupils during the follow-up period was also recorded. In our study, the IOP of the patients after surgery was measured every day after cornea transplantation. We used atropine or tropicamide eye drops for the patients who had dramatic inflammation, usually for one or three days for atropine (qd) or one week for tropicamide (qid). If a wide, fixed pupil was recorded 3 weeks after the final use of atropine or tropi-camide, some of the patients received pilocarpine eye drops (qid) for one or two weeks.

Incidence differences were statistically analyzed using Fisher’s exact test and the chi-square test using SPSS19.0 software (IBM SPSS Statistics, New York, NY, USA). *p*-values of <0.05 were considered statistically significant.

## 3. Results

### 3.1. Causes of Pupillary Abnormalities after PKP

A total of 152 cases with follow-up time of more than 3 months were included in this study; 93 (61.18%) of these cases had postoperative pupillary abnormalities, including pupil morphology and size abnormalities. Seventy (75.27%) cases of pupillary abnormalities had a clear etiology, including preoperative uveitis, traumatic iris defects, preoperative pupillary abnormalities, gonioplasty, iridocorneal endothelial syndrome, and excision of iris cysts. None of the 93 cases had fixed, dilated pupils before surgery. The etiology of the postoperative pupillary abnormalities in the remaining 23 patients who had normal pupils preoperatively was unclear (Table 1).

Some of the patients received pilocarpine eye drops (qid) for one or two weeks, without any change of the pupil during the observed period.

### 3.2. Primary Diseases in UZS Patients

The incidence rate of UZS was 15.13% (23/152), including eight cases of keratoconus (34.78%), six cases of viral keratitis (26.09%), four cases of leukoma (17.39%), two cases of fungal corneal ulcer (8.70%), two cases of corneal endothelial decompensation (8.69%), and one case of corneal degeneration (4.35%) (Table 2).

Fisher’s exact test indicated that the incidence of UZS significantly differed among these groups (*p* = 0.011, a = 0.05); however, the subsequent comparison between any two groups indicated that only the difference between the keratoconus group and fungal corneal ulcer group was statistically significant (*p* = 0.003, a = 0.0032). Patients with keratoconus were more likely to develop UZS after PKP.

### 3.3. IOP and UZS after PKP in Keratoconus Patients

Because keratoconus patients accounted for the largest proportion to develop UZS after PKP in our study, the relationship between IOP and UZS development after PKP was investigated using keratoconus patients as an example. Among the 152 patients who underwent PKP, 19 had keratoconus. Among them, eight (42.11%) developed UZS, and six had elevated IOP after surgery (Table 3). No significant difference was observed in the incidence of UZS between patients with and without a history of elevated IOP after PKP (*p* = 0.319).

### 3.4. Spontaneous Recovery of Pupils in Patients Who Developed UZS after PKP

Twenty-one of the 23 patients with UZS were followed up for >6 months. The anterior segment photographs revealed that seven patients’ (33.33%) pupils spontaneously recovered to their preoperative level without symptomatic treatment. Three patients (42.86%) recovered within 4 months, and four patients (57.14%) recovered after 6 months. The shortest recovery time was 48 days, and the longest was 1.5 years. Of the seven UZS patients who spontaneously recovered, three had keratoconus, one had leukoma, one had viral keratitis, one had fungal corneal ulcer, and one had corneal endothelial decompensation (Table 4). Photographs of three patients before and after PKP are provided in Figure 1, Figure 2 and Figure 3, and photographs of the pupils of a patient whose primary disease was fungal corneal ulcer recovered to normal pupils are shown in Figure 4.

## 4. Discussion

The incidence of UZS after PKP in this study was 15.13%, which is consistent with the 2.2–17.7% incidence reported in the literature [12]. The primary diseases of these UZS patients included keratoconus, viral keratitis, leukoma, fungal corneal ulcer, corneal endothelial decompensation, and corneal degeneration; among these, keratoconus accounted for the highest percentage (34.78%). Statistical analysis revealed that the incidence of postoperative UZS in patients with keratoconus was higher than that in patients with fungal corneal ulcer (42.11% vs. 6.25%, respectively).

Although the etiology of UZS has not yet been determined, various mechanisms have been proposed. Urrets-Zavali [1] believes that the use of atropine during surgery causes the iris to make contact with the peripheral cornea, leading to peripheral anterior synechiae and secondary glaucoma. However, a prospective study by Geyer et al. [18] in 1991 concluded that the occurrence of UZS after keratoplasty was not associated with the use of atropine.

Presently, it is generally believed that iris ischemia is the main cause of UZS [1,11,13,14,15]. Davies et al. [14] believe that, after the elevated IOP presses the iris against the back of the cornea, the sphincter pupillae and iris will be ischemic due to the occlusion of the greater arterial circle of the iris, leading to sphincter paralysis. The degree of ischemia and iris atrophy determines the degree of pupil dilation and whether permanent or reversible dilation occurs. It should be noted that the above mechanism is more likely to occur in patients with keratoconus, because the iris abnormality and ocular rigidity are lower than normal in such patients. The direct trauma caused by corneal scissors to the iris is also one of the mechanisms that cause iris ischemia and subsequent tonic pupil.

In 1995, Tuft [19] demonstrated the presence of iris ischemia in UZS patients after PKP for keratoconus using anterior-segment fluorescein angiography. He suggested that UZS occurred due to iris-ischemia and sphincter-pupillae damage as a result of ischemia of the iris root vessels, caused by elevated postoperative IOP. Additionally, he specifically proposed not to leave viscoelastic agents in the anterior chamber, which could reduce the outflow of aqueous humor after surgery. Many studies have suggested that the residual viscoelastic agents in the anterior chamber are toxic to the sphincter pupillae or blood vessels, which may increase the incidence of UZS [2,3]. However, others have suggested that the use of viscoelastic agents may not have much impact on the occurrence of UZS, because no iris atrophy has been observed during cataract surgery [5] wherein viscoelastic agents are widely used. It is also notable that leaving gas bubbles or injecting air into the anterior chamber of a phakic eye can cause pupillary blockage and increased IOP, leading to the occurrence of UZS [6,13]. Some believe that, although mannitol is administered before surgery, the sudden increase in IOP due to the formation of large gas bubbles may offset the action of mannitol and cause iris ischemia [20].

Most literature suggests that the occurrence of UZS is associated with elevated IOP [1,2,14,19]. Although no statistically significant difference was found in the incidence of UZS between keratoconus patients with and without postoperative transient high IOP in our study, the absolute difference between these two groups was more than doubled (4/6: 4/13). Significant differences are likely between these two groups when the sample size is increased. Transient high IOP may increase the occurrence of UZS after PKP for keratoconus.

A study of UZS after PKP for keratoconus found that approximately two-thirds of patients exhibited some form of pupillary response in the first few weeks after surgery, or even a return to normal pupil size [15]. Seven patients with UZS in our study recovered spontaneously after surgery, indicating that some UZS patients can recover spontaneously. The shortest recovery time in this study was 48 days. As mentioned earlier, the degree of ischemia and iris atrophy determines the extent of pupil dilation and whether it is permanent or reversible.

Sharif and Casey [21] reviewed the data of 100 patients who underwent PKP for keratoconus between 1968 and 1986. They concluded that the incidence of UZS decreased from 4% to 1.5% after routine intravenous injection of 20% mannitol. The hypertonic solution reduced the vitreous volume and the extent to which the iris was compressed [5].

UZS causes pain and inconvenience to patients, particularly young patients. Glare and appearance problems may occur throughout the lifespan of the patients. The current treatment includes wearing sunglasses or colored contact lenses. Clinicians should focus on keratoconus, viral keratitis, leukoma, fungal corneal ulcer, corneal endothelial decompensation, and corneal degeneration before PKP surgery. Postoperative observation should also be strengthened, particularly with regard to monitoring of perioperative IOP; patients with elevated IOP should be administered treatment in a timely manner. An intravenous infusion of 20% mannitol should be routinely performed after surgery. Iris damage by trephine and scissors should be avoided during surgery to reduce the incidence of this complication.

## 5. Conclusions

The incidence of UZS after PKP in this study was 15.13%. Patients with keratoconus may be more prone to UZS, and postoperative transient high IOP may increase the risk of UZS after PKP for keratoconus.

## Figures and Tables

**Figure 1 jcm-11-01175-f001:**
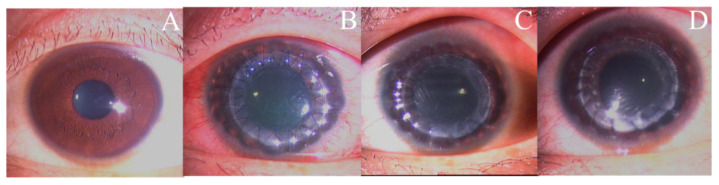
Anterior segment photographs of three UZS patients during the 3-month follow-up period. A 21-year-old male with keratoconus in his left eye: (**A**) preoperatively; (**B**) three days postoperatively; (**C**) one month postoperatively; (**D**) three months postoperatively.

**Figure 2 jcm-11-01175-f002:**
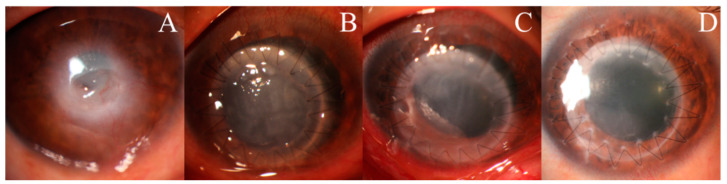
Anterior segment photographs of a UZS patient during the 7-month follow-up period. A 40-year-old male with viral keratitis in his right eye: (**A**) preoperatively; (**B**) four days postoperatively; (**C**) one month postoperatively; (**D**) seven months postoperatively.

**Figure 3 jcm-11-01175-f003:**
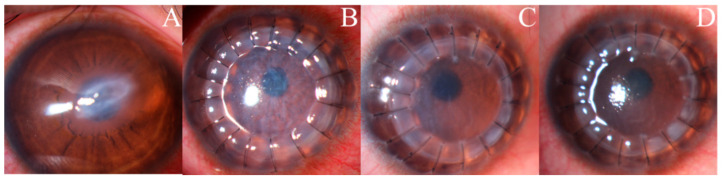
Anterior segment photographs of a patient with a normal pupil during the 4.5-month follow-up period. A 32-year-old male with keratoconus (with acute hydrops) in his right eye: (**A**) preoperatively; (**B**) one day postoperatively; (**C**) two months and two weeks postoperatively; (**D**) four months and two weeks postoperatively.

**Figure 4 jcm-11-01175-f004:**
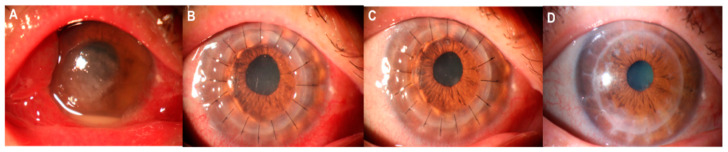
Anterior segment photographs of a UZS patient during the 15-month follow-up period: (**A**) preoperatively; (**B**) fourteen days postoperatively; (**C**) one and half months postoperatively; (**D**) one year and three months postoperatively.

**Table 1 jcm-11-01175-t001:** Etiology of the 93 patients with pupillary abnormalities after PKP.

Cause	Number of Patients	Percentage (%)
Uveitis	47	50.54
Trauma	10	10.75
Preoperative papillary abnormalities	9	9.68
Gonioplasty	2	2.15
Iridocorneal endothelial syndrome	1	1.08
Excision of iris cyst	1	1.07
Unclear etiology *	23	24.73
Total	93	100.00

* The preoperative pupil was normal. The etiology of the pupil abnormality was unclear during and after surgery.

**Table 2 jcm-11-01175-t002:** Incidence of UZS after PKP in patients with different primary diseases.

Primary Disease	Number of UZS Patients	Total Number of Patients	Percentage * (%)	Incidence (%)
Keratoconus	8	19	34.78	42.11
Viral keratitis	6	16	26.08	37.50
Leukoma	4	30	17.39	13.33
Fungal corneal ulcer	2	32	8.70	6.25
Corneal endothelial decompensation	2	9	8.70	22.22
Corneal degeneration	1	3	4.35	33.33

UZS—Urrets-Zavalia syndrome. * The percentage is the ratio of the number of postoperative UZS patients with different primary diseases to the total number of postoperative UZS patients (23 cases); Fisher’s exact test produced *p* = 0.011.

**Table 3 jcm-11-01175-t003:** Incidence of UZS in patients with elevated IOP after PKP.

IOP after PKP	Number of UZS Patients	Total Number of Patients	Incidence (%)
With a history of increased IOP	4	6	66.67
Without a history of increased IOP	4	13	30.77

UZS—Urrets-Zavalia syndrome; IOP—intraocular pressure; PKP—penetrating keratoplasty. Fisher’s exact test yielded *p* = 0.319.

**Table 4 jcm-11-01175-t004:** Spontaneous recovery from pupil abnormalities in patients who developed UZS after PKP.

Primary Disease	Number of UZS Patients	Number of Patients Who Spontaneously Recovered	Incidence * (%)
Keratoconus	8	3	37.50
Viral keratitis	4	1	25.00
Leukoma	4	1	25.00
Fungal corneal ulcer	2	1	50.00
Corneal endothelial decompensation	2	1	50.00
Corneal degeneration	1	0	-

UZS—Urrets-Zavalia syndrome. * The incidence is the ratio of the number of patients who spontaneously recovered to the total number of UZS patients with different primary diseases.

## Data Availability

All data generated or used during the study appear in the submitted article.

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
