# Peer review of "Risk Factors of Urrets-Zavalia Syndrome after Penetrating Keratoplasty"

_jcm, 2022, doi:10.3390/jcm11051175_

Round 1

Reviewer 1 Report

In the current work entitled" Risk Factors of Urrets-Zavalia Syndrome After Penetrating Keratoplasty"  the authors described that postoperative transient high IOP may increase the incidence of UZS after PKP in patients with keratoconus, and consequently more attention should be afforded to these
patients to prevent the occurrence of UZS after PKP.

minor comments:

  1. Authors should define better "USZ" in the introduction
  2. The last sentence "In our study, 33.33% patients with UZS who were followed-up for >6 months recovered spontaneously, indicating that spontaneous recovery is possible", is an observation but not a conclusion and should be removed 

Author Response

Response to Reviewer 1 Comments

Point 1: Authors should define better "USZ" in the introduction

Response 1: Thanks for your advice. We have added further definitions of USZ in the introduction section(In red)

Point 2: The last sentence "In our study, 33.33% patients with UZS who were followed-up for >6 months recovered spontaneously, indicating that spontaneous recovery is possible", is an observation but not a conclusion and should be removed 

Response 2: Your suggestion is quite correct

Reviewer 2 Report

Thanks for allowing me to review this article. 

Trying to find risk factors associated with UZS is interesting. 

The study it self is well presented. 

I have a few reservations about the study:

The main issues are:

1) The criteria for diagnosis of UZS are not defined in the study. UZS is mainly used for description of a wide, fixed pupil post-PK surgery. The picture (figure 4) shows a mid-dilated pupil, that is very much similar to the pre-operative state. It raises the question of what were the criteria that were used , and possibly that the authors are reporting on post-PK pupillary abnormalities, not on UZS. 

Additionally, in the Methods section (line 81) the authors state that for cloudy corneas the pupil diameter pre-PK was measured after removal of the corneal button. This can easily mask results, as anesthesia, topical drugs and the removal of the corneal button and 'open-sky' situation changes the pupillary dimensions.

2) Measuring the

The report deals with patients from 2014-2016, and the criteria is FU of over 3 months. What happened 2016-2022? or at least 2016-2020?

Expanding the time reviewed will make the N higher, and will make finding true associations more likely. With such small N in several of the groups, coming to any significant conclusion is very difficult. 

3) The authors don't report on any intra-operative detail. What was the diameter of the corneal graft? what was the diameter of the recipient cornea? how many sutures? how long did the surgery take?

The lack of standardization makes drawing any conclusion from the data very difficult. 

Other smaller issues are:

1) In the introduction the authors claim that factors affecting were described (lines 45-47). However, no references is given. It seems like UZS has been described in patients with the listed conditions, rather than these conditions affecting UZS. 

2) In the introduction (lines 53-67) the authors try to explain why this is an important topic for central-china cities. I didn't understand their logic as to why this would be more important for central-china than to developed and poor countries. This should be either changed or discarded. Also, if claiming that, they should provide data to support the claim (for example: more patients question about UZS?, there is less availability to manage complications? the number of ophthalmologists per 100,000 population is different?). 

2) the authors report that many patients were using atropine and other pupil-affecting drugs after surgery, and that the diagnosis was made 3 weeks after stopping these, however no data is presented as to when was that relative to the time of original surgery (did they use atropine for 1 week? 1 month? 3 months?; did the frequency of application changed?). 

3) IOP elevation is not clearly defined (any occurrence of IOP > 21?; on which day post-operatively?), so breaking the KC group into two is difficult to follow, and bears little information to the reader and little clinical relevance

4) The authors report on recovery from UZS, but only share the time period. 

Were the patients given any treatment? topical steroids, pilocarpine? when were the sutures removed and did it have any relation to resolution?

5) line 224 - This is unsupported. Getting a larger sample size is a great idea. The difference might be bigger or non-existent. That is why one might want to get a larger sample - because we don't know what the result will be. 

6) line 231 - Some authors POSTULATE that ischemia and atrophy are the causes of UZS, that had not been proved, so recovery might be related to these factors, but this is not certain. 

7) line 239-240 - There are a few more treatment options for wide fixed pupils.

8) line 241 - What does it mean "clinicians should focus on ...before PKP surgery"? 

9) line 245 - this is a completely unfounded claim. The authors did not evaluate wether mannitol help with reducing UZS. 

10) line 246 - Again, while might be true, this claim is not supported by the authors data. 

11) in the conclusion (line 250) The data did NOT show a difference between elevated IOP and UZS. 

12) line 251-252: What does "greater care should be afforded" mean?

The study did not look at ways to decrease the rate of UZS, and has no relevant suggestions on how to do so. It reports on the incidence of UZS in a past cohort. 

Author Response

Response to Reviewer 2 Comments

The main issues are:

1) The criteria for diagnosis of UZS are not defined in the study. UZS is mainly used for description of a wide, fixed pupil post-PK surgery. The picture (figure 4) shows a mid-dilated pupil, that is very much similar to the pre-operative state. It raises the question of what were the criteria that were used , and possibly that the authors are reporting on post-PK pupillary abnormalities, not on UZS. 

Additionally, in the Methods section (line 81) the authors state that for cloudy corneas the pupil diameter pre-PK was measured after removal of the corneal button. This can easily mask results, as anesthesia, topical drugs and the removal of the corneal button and 'open-sky' situation changes the pupillary dimensions.

Response : Thank you for your suggestion. We have modified this part.(please see line 72-74 in red). As for the measurement method of pupil diameter, we agree with you. In addition, there is no other effective method to measure pupil diameter due to shield of lesions before surgery, intraoperative measurement is more intuitive. It is worth mentioning that UZS was diagnosed only if a wide, fixed pupil were recorded 3 weeks discontinued the application of topicamide.

2) Measuring the The report deals with patients from 2014-2016, and the criteria is FU of over 3 months. What happened 2016-2022? or at least 2016-2020?

Response : Thank you for your suggestion. We only sorted out the cases from 2014 to 2016. Due to the busy clinical work, we have not sorted out the cases from 2016 to 2022. We plan to continue to analyze the cases from 2016 to 2022 with a larger sample size after this article.

3) The authors don't report on any intra-operative detail. What was the diameter of the corneal graft? what was the diameter of the recipient cornea? how many sutures? how long did the surgery take?

The lack of standardization makes drawing any conclusion from the data very difficult. 

Response :Thank you for your suggestion. We have modified this part.(please see line74-76 in red)

Other smaller issues are:

1) In the introduction the authors claim that factors affecting were described (lines 45-47). However, no references is given. It seems like UZS has been described in patients with the listed conditions, rather than these conditions affecting UZS. 

Response : Thank you for your suggestion. we have modified this part and added relevant references.(please see line 44-45 in red)

2) In the introduction (lines 53-67) the authors try to explain why this is an important topic for central-china cities. I didn't understand their logic as to why this would be more important for central-china than to developed and poor countries. This should be either changed or discarded. Also, if claiming that, they should provide data to support the claim (for example: more patients question about UZS?, there is less availability to manage complications? the number of ophthalmologists per 100,000 population is different?). 

Response : Thank you for your suggestion. we think we did not express our meaning correctly. We meant that UZS is becoming more and more important in developing countries with rapid development, especially in centrol China where is plain and relatively densely populated.  Because people's living standard is gradually improving , of course, this can also be said to be the whole of China, we have made the corresponding modifications,thank you

2) the authors report that many patients were using atropine and other pupil-affecting drugs after surgery, and that the diagnosis was made 3 weeks after stopping these, however no data is presented as to when was that relative to the time of original surgery (did they use atropine for 1 week? 1 month? 3 months?; did the frequency of application changed?). 

Response : We used atropine or tropicamide eye drops for the patients that have dramatic inflammation, usually one or three days for atropine (qd) or one week for tropicamide(qid). We added the data in the manuscript accordingly(please see line 83-86 in red)

3) IOP elevation is not clearly defined (any occurrence of IOP > 21?; on which day post-operatively?), so breaking the KC group into two is difficult to follow, and bears little information to the reader and little clinical relevance

Response : Thank you for your suggestion. We have added this part. (please see line 87-88 in red)

4) The authors report on recovery from UZS, but only share the time period. 

Were the patients given any treatment? topical steroids, pilocarpine? when were the sutures removed and did it have any relation to resolution?

Response : topical steroids but not pilocarpine is used after surgery in our patients. We usually remove the sutures one year after surgery, which changed the visual acuity. (data not shown)

5) line 224 - This is unsupported. Getting a larger sample size is a great idea. The difference might be bigger or non-existent. That is why one might want to get a larger sample - because we don't know what the result will be. 

Response : We totally agree with you, thank you

6) line 231 - Some authors POSTULATE that ischemia and atrophy are the causes of UZS, that had not been proved, so recovery might be related to these factors, but this is not certain. 

Response : Ischemia and atrophy are the causes of UZS, but had not been proved, we agree with you, However, we think ischemia and atrophy are one of the main factors affecting UZS recovery

7) line 239-240 - There are a few more treatment options for wide fixed pupils.

Response : pilocarpine has been reported previously by other scientists, and surgery that suture the pupil smaller could be used if further cataract or other inner eye surgery are going to be carried out. 

8) line 241 - What does it mean "clinicians should focus on ...before PKP surgery"? 

Response : Clinicians should be concerned for keratoconus, viral keratitis, leukoplakia, fungal corneal ulcer, corneal endothelial decompensation, and corneal degeneration before PKP surgery. If these conditions are the cause, clinicians should consider the possibility of postoperative UZS

9) line 245 - this is a completely unfounded claim. The authors did not evaluate wether mannitol help with reducing UZS. 

Response : Routine intravenous infusion of 20% mannitol is the routine treatment of PKP after surgery to prevent IOP elevation, and we think the effect is positive, which is helpful to reduce intraocular pressure and other complications caused by surgery.

10) line 246 - Again, while might be true, this claim is not supported by the authors data. 

Response : It is true that our data cannot reach this conclusion, but we think that the injury of surgical instruments such as trephine and scissors should be avoided as much as possible. we write it here to remind readers that they should pay attention to these problems

11) in the conclusion (line 250) The data did NOT show a difference between elevated IOP and UZS. 

Response : Yes, although it is expected that elevated IOP could do harm for the pupil, our data did not show a relevance between elevated IOP and UZS

12) line 251-252: What does "greater care should be afforded" mean?

Response : we rewrite it with “more attention should be paid”

Reviewer 3 Report

The paper is interesting and well written. I feel some minor English editing is necessary (e.g. of the following sentence: "It's worth mentioning that those patients with who keratoconus, bullous keratitis and 79
other conditions that the cornea is not so cloudy, pupil diameter could be measured before 80
surgery, while for those the cornea is so cloudy that the pupil could not be measured be- 81
fore surgery, we measure them after removing the corneal buttons")

Moerover I would like to ask why visual acuity was not considered in the statistics since it is a major finding of all our ophthalmic surgery, although I understand it was not in the purpose of the article: please specify this in the text

Author Response

Response to Reviewer 3 Comments

Point 1: The paper is interesting and well written. I feel some minor English editing is necessary (e.g. of the following sentence: "It's worth mentioning that those patients with who keratoconus, bullous keratitis and other conditions that the cornea is not so cloudy, pupil diameter could be measured before surgery, while for those the cornea is so cloudy that the pupil could not be measured before surgery, we measure them after removing the corneal buttons")

Response 1: Thank you for your affirmation. We have further optimized the relevant language for the full text.

Point 2: Moerover I would like to ask why visual acuity was not considered in the statistics since it is a major finding of all our ophthalmic surgery, although I understand it was not in the purpose of the article: please specify this in the text

Response 2: Thank you for your advice. We would like to report the influence of UZS on visual acuity in the next manuscript, it is a major finding of all our ophthalmic surgery.

Round 2

Reviewer 2 Report

Thank you for the opportunity to review your work. It is obvious that a lot of work has been invested in the article.  

While I think that several issues were addressed, I don't find them all to be resolved.

I am still uncertain about the actual diagnosis of UZS, as some patients seem to merely have post-operative pupillary abnormalities rather than UZS. This makes the analysis of data questionable. 

The authors describe collecting many data points, but not describing it in the results, which could maybe shed some light on that (for example mean pupil diameter before / after and amount of change in dilation). Some of the data that was collected could be confounding the results, but that was not explored (difference in incidence for different surgical parameters). 

The post-op treatment and post-UZS diagnosis was also not standardized making the conclusions even weaker. 

See previous review for more detailed breakdown. 

My apologies if you think I have been too critical of your work. I know you have done a lot of gathering of details and chart reviewing. You have simply left out too much important information and made statements that are simply not supported. I appreciate this was not intentional.

Author Response

Thank you for your advice and comments, we try to interpret them below,

1.I am still uncertain about the actual diagnosis of UZS, as some patients seem to merely have post-operative pupillary abnormalities rather than UZS. This makes the analysis of data questionable. 

Response1 : We added the diagnosis standard for UZS according the references cited in this manuscript in the INTRODUCTION (line 33-35) as “ it has been defined as a fixed wide pupil after intraocular surgeries without the affecting of pupil dilation medicine including atropine and topicamide or clear etiology, including preoperative uveitis, traumatic iris defects, preoperative pupillary abnormalities, gonioplasty, iridocorneal endothelial syndrome, and excision of iris cysts et al.”, and interpreted in the Materials and Methods(line 62-64 )that “If a wide, fixed pupil were recorded 3 weeks after the terminate use of atropine or topicamide, excluding clear etiology for pupil dilation, including preoperative uveitis, traumatic iris defects, preoperative pupillary abnormalities, gonioplasty, iridocorneal endothelial syndrome, and excision of iris cysts et al.the diagnosis of postoperative UZS was made in this study.

2.The authors describe collecting many data points, but not describing it in the results, which could maybe shed some light on that (for example mean pupil diameter before / after and amount of change in dilation). Some of the data that was collected could be confounding the results, but that was not explored (difference in incidence for different surgical parameters). 

Response2 : A total of 152 cases with following-up time more than 3 months were included in this study. Because of the uncertainty of the patient's visit, their postoperative data were not necessarily at the same time, but all of them have a following-up time more than 3 months, to make sure that all of them have time to be excluded from pupil dilation caused by medicine including atropine and topicamide

3.The post-op treatment and post-UZS diagnosis was also not standardized making the conclusions even weaker. 

Response3 : The first reason that make the treatment difficult to be standardized is the retrospective but not prospective design study for this manuscript; the second reason is that the patients have different primary disease for PKP surgery, thus need different medicine after surgery. For those we diagnosed UZS, some received pilocarpine but have no effect in reducing the extent of UZS. (line 76,88-89)